# Identification of Growth Factors, Cytokines and Mediators Regulated by *Artemisia annua* L. Polyphenols (pKAL) in HCT116 Colorectal Cancer Cells: TGF-β1 and NGF-β Attenuate pKAL-Induced Anticancer Effects via NF-κB p65 Upregulation

**DOI:** 10.3390/ijms23031598

**Published:** 2022-01-29

**Authors:** Eun Joo Jung, Anjugam Paramanantham, Hye Jung Kim, Sung Chul Shin, Gon Sup Kim, Jin-Myung Jung, Soon Chan Hong, Ky Hyun Chung, Choong Won Kim, Won Sup Lee

**Affiliations:** 1Department of Internal Medicine, Institute of Health Sciences, Gyeongsang National University Hospital, Gyeongsang National University College of Medicine, 15 Jinju-daero 816beon-gil, Jinju 52727, Korea; eunjoojung@gnu.ac.kr (E.J.J.); anju.udhay@gmail.com (A.P.); 2Research Institute of Life Science, College of Veterinary Medicine, Gyeongsang National University, Jinju 52828, Korea; gonskim@gnu.ac.kr; 3Department of Pharmacology, Institute of Health Sciences, Gyeongsang National University College of Medicine, Jinju 52727, Korea; hyejungkim@gnu.ac.kr; 4Department of Chemistry, Research Institute of Life Science, Gyeongsang National University, Jinju 52828, Korea; sshin@gnu.ac.kr; 5Department of Neurosurgery, Institute of Health Sciences, Gyeongsang National University Hospital, Gyeongsang National University College of Medicine, Jinju 52727, Korea; gnuhjjm@gnu.ac.kr; 6Department of Surgery, Institute of Health Sciences, Gyeongsang National University Hospital, Gyeongsang National University College of Medicine, Jinju 52727, Korea; hongsc@gnu.ac.kr; 7Department of Urology, Institute of Health Sciences, Gyeongsang National University Hospital, Gyeongsang National University College of Medicine, Jinju 52727, Korea; kychung@gnu.ac.kr; 8Department of Biochemistry, Institute of Health Sciences, Gyeongsang National University College of Medicine, Jinju 52727, Korea; cwkim@gnu.ac.kr

**Keywords:** *Artemisia annua* L. polyphenols, anticancer effect, colorectal cancer, antibody array, cytokine, TGF-β1, NGF-β, NF-κB p65

## Abstract

The anticancer effects of natural phytochemicals are relevant to the modulation of cytokine signaling pathways in various cancer cells with stem-like properties as well as immune cells. The aim of this study was to elucidate a novel anticancer mechanism of *Artemisia annua* L. polyphenols (pKAL) involved in the regulation of growth factors, cytokines and mediators in stem-like HCT116 colorectal cancer cells. Through RayBiotech human L-1000 antibody array and bioinformatics analysis, we show here that pKAL-induced anticancer effects are associated with downregulation of growth factor and cytokine signaling proteins including TGFA, FGF16, PDGFC, CCL28, CXCR3, IRF6 and SMAD1. Notably, we found that TGF-β signaling proteins such as GDF10, ENG and TGFBR2 and well-known survival proteins such as NGF-β, VEGFD and insulin were significantly upregulated by pKAL. Moreover, the results of hematoxylin staining, cell viability assay and Western blot analysis demonstrated that TGF-β1 and NGF-β attenuated pKAL-induced anticancer effects by inhibiting pKAL-induced downregulation of caspase-8, NF-κB p65 and cyclin D1. These results suggest that certain survival mediators may be activated by pKAL through the TGF-β1 and NGF-β signaling pathways during pKAL-induced cell death and thus, strategies to inhibit the survival signaling are inevitably required for more effective anticancer effects of pKAL.

## 1. Introduction

Colorectal cancer is one of the leading diseases with high mortality and morbidity in both men and woman worldwide. It is known that about 50% of colorectal cancer patients experience anticancer drug resistance, metastasis and tumor recurrence related to the modulation of colorectal cancer stem cells, which have been considered as a major therapeutic target for colorectal cancer [1,2,3]. Colorectal cancer phenotypes can be classified according to their correlation with colon-crypt location and Wnt signaling as follows: stem-like subtype, transit-amplifying subtype, goblet-like subtype and enterocyte subtype [4]. HCT116 colorectal cancer cells belong to a stem-like subtype and express stem cell marker proteins such as CD24, CD44 and CD133 on the cell surface [3,5]. Stem-like properties of HCT116 cells were enhanced by the induction system of GATA6 overexpression in HCT116 cells, resulting in increased CD44 and CD133 in conditioned media collected from stemness-high HCT116 cells [6]. Notably, elevated levels of vascular endothelial growth factor A (VEGF-A) and interleukin-8 (IL-8) through activation of the EGFR/AKT/NF-κB pathway were detected in the conditioned media of stemness-high HCT116 cells [6]. In addition, the IL-8 (also known as CXCL8) mediated CXCR1/2 signaling pathway has been shown to be in involved in chemoresistance induced by 5-FU in HCT116 cells [7]. These findings suggest that a variety of growth factors and cytokines may be modulated under certain stress circumstances in stem-like HCT116 cells and may be involved in the regulation of cancer cell survival, proliferation, metastasis and chemotherapeutic drug resistance.

Polyphenols are phytochemicals that have a phenolic structure and are present in large amounts in various foods such as fruits, vegetables, nuts, spices, coffee, tea, red wine and olive oil. Phenolic compounds can be classified according to their chemical structures as follows: hydroxycinnamic acids (e.g., ferulic acid, caffeic acid), hydroxybenzoic acids (e.g., gallic acid), stilbenes (e.g., resveratrol) and flavonoids including anthocyanidins (e.g., cyanidin), flavanols (e.g., catechins), flavanones (e.g., naringenin), flavonols (e.g., quercetin, kaempferol), flavones (e.g., apigenin, luteolin) and isoflavones (e.g., genistein) [8]. Bioactive natural plant polyphenols are known to be very beneficial to human health due to their antioxidant, anti-inflammatory, antidiabetic, anti-neurodegenerative and anti-cancer effects [9,10]. In particular, dietary polyphenols exhibit effective and safer anticancer effects and can be considered as a new alternative therapy for certain patients suffering from various side effects caused by conventional chemotherapy and radiation therapy [10]. Polyphenols play an important role in inhibition of cell survival and the anticancer mechanisms by olive oil polyphenols are known as follows: induction of cell cycle arrest via upregulation of p21 and downregulation of cyclin-dependent kinase 1 (CDK1); induction of apoptosis via upregulation of caspase-3/-7, p53, Bak and Bax; inhibition of cell viability and proliferation; inhibition of angiogenesis via downregulation of VEGF; inhibition of inflammation via tumor necrosis factor alpha (TNF-α), IL-6 and nuclear factor kappa B (NF-κB) [9]. However, certain types of polyphenols, such as luteolin and hydroxycinnamic acid derivatives, are known to play a role in neuronal survival and rescue from neurodegeneration by controlling apoptotic factors and neuroprotective neurotrophic factors (NTFs) such as nerve growth factor (NGF), brain-derived neurotrophic factor (BDNF), NT4/5 and NT3 [8,11]. These studies indicate that polyphenols have a role in cell death or survival depending on their chemical structures, cell/tumor types and intracellular/extracellular signaling targets.

We previously reported that polyphenols extracted from *Artemisia annua* L. (pKAL) had ROS-independent cell death in HCT116 colorectal cancer cells through activation of the p53-dependent apoptotic signaling pathways [12,13]. In this study, we show that pKAL can induce intracellular and extracellular vesicles in HCT 116 cells, suggesting a potential survival role for pKAL associated with vesicle accumulation during pKAL-induced cell death. To better understand the pKAL-induced cell death mechanism, we identified 23 significant pKAL-regulated growth factors, cytokines and signaling mediators through RayBiotech Human L-1000 antibody array analysis in stem-like HCT 116 cells. Furthermore, our study reveals that pKAL can induce cell survival signaling pathways to some extent through activation of certain types of growth factors and cytokines such as TGF-β1 and NGF-β under a massive pKAL-induced cell death circumstance. These findings suggest that pKAL plays a dual role by inducing both cell survival and death signaling pathways and other drugs for combination treatment with pKAL to inhibit pKAL-induced survival signaling pathways are necessarily required for more effective pKAL-induced anticancer effects in colorectal cancer therapy.

## 2. Results

### 2.1. pKAL Induces Intracelluar and Extracelluar Vesicles in HCT116 Cells

To further elucidate pKAL-induced anticancer mechanisms in HCT116 cells, the long-term effects of pKAL on the regulation of cell viability were analyzed by CCK-8 assay at 60 and 84 h after pKAL treatment. Unexpectedly, the ability of pKAL to reduce cell viability was somewhat reduced at 84 h than 60 h pKAL treatment: compare 60 h (64%) and 84 h (88%) at 25 μg/mL pKAL treatment; compare 60 h (21%) and 84 h (26%) at 50 μg/mL pKAL treatment, suggesting an induction of survival pathway during long-term treatment of pKAL (Figure 1a). To better understand this phenomenon, we investigated the morphological changes caused by the long-term effect of pKAL by phase-contrast microscopy and hematoxylin staining in HCT116 cells. As shown in Figure 1b, cell morphology was significantly altered to various enlarged shapes or small round shapes by 50 μg/mL pKAL treatment for 60 h (compare panels a and b). Moreover, certain vesicles were found in the intracellular region (representatively indicated by arrow No. 1 in panel b’) and in the extracellular region near the plasma membrane (indicated by arrows No. 2 and 3 in panel b’) (Figure 1b). The morphological changes by pKAL were further confirmed by phase-contrast microscopy after hematoxylin staining. As shown in Figure 1c, pKAL altered cell morphology into various small round or enlarged cells: small round cells were more strongly stained by hematoxylin solution than enlarged cells (compare panels a and b). In particular, a pKAL-induced intracellular vesicle was clearly detected in the periphery of the nuclear membrane of enlarged cells (indicated by arrow No. 1 in panel b’) and various types of small round cells or extracellular vesicles induced by pKAL were strongly stained by hematoxylin solution (indicated by arrows No. 2–7 in panel b’) (Figure 1c). These results suggest that the intracellular and extracellular vesicles induced by pKAL may be associated with pKAL-induced survival pathways during pKAL-induced cell death.

### 2.2. Identification of Growth Factors, Cytokines and Signaling Mediators Regulated by pKAL

It is known that extracellular vesicles can be secreted or shed by various cancer cells and colorectal cancer-derived small extracellular vesicles are targeted to liver tissue and induce an inflammatory premetastatic niche in liver metastasis [14,15]. To identify unknown growth factors, cytokines and signaling mediators regulated by pKAL, we performed proteomic analysis in stem-like HCT116 cells using the RayBiotech Human L-1000 antibody array. As a result, 10 growth factor and cytokine signaling proteins were upregulated by pKAL more than 1.4-fold and their gene symbols are as follows: INS, ENG, GDF10, TNFSF15, KRT8, SIGIRR, NGF, MMP24, VEGFD and TGFBR2 (Table 1). In contrast, 13 growth factor and cytokine signaling proteins were downregulated by pKAL more than 1.4-fold and their gene symbols are as follows: IRF6, IL19, FGF16, TGFA, CTGF, CCL28, MMP15, BPIFA1, NTF4, PDGFC, CXCR3, GMNN and SMAD1 (Table 2).

### 2.3. Bioinformatics Analysis

#### 2.3.1. Expression Heatmap and Keywords for pKAL-Regulated Proteins

To visualize expression data for pKAL-regulated proteins, heatmap analysis was performed using Excel based Differentially Expressed Gene Analysis (ExDEGA) software (Figure 2a). In addition, keywords and functional information for pKAL-regulated proteins were analyzed from the DAVID database and the UniProt website (Figure 2b). The results of bioinformatics analysis suggested that pKAL-induced anticancer effects are significantly associated with downregulation of growth factors (FGF16, TGFA, NTF4, PDGFC), cytokines (CCL28, IL19) and other signaling mediators (GMNN, SMAD1, CTGF, MMP15, CXCR3, BRIFA1, IRF6). However, pKAL also significantly upregulated TGF-β signaling proteins such as endoglin (ENG), growth/differentiation factor 10 (GDF10) and TGF-beta receptor type-2 (TGFBR2) and well-known survival proteins such as insulin (INS), NGF-β (NGF) and vascular endothelial growth factor D (VEGFD). These findings suggest that pKAL may have a dual role by regulating both cell survival and death signaling.

#### 2.3.2. Gene Category Classification for pKAL-Regulated Proteins

We analyzed gene ontology for pKAL-regulated proteins using ExDEGA software and the DAVID database and the resulting gene category chart is shown in Figure 3. The results suggested that pKAL may have a dual role by modulation of the TGF-beta receptor signaling pathway through upregulation of GDF10, ENG and TGFBR2 and downregulation of SMAD1. In addition, the results suggested that pKAL-induced anticancer effects could be inhibited by upregulation of NGF-β, insulin and VEGFD classified into the intracellular vesicle category in Figure 3.

#### 2.3.3. Protein–Protein Interaction Network for pKAL-Regulated Proteins

As shown in Figure 4, we analyzed the protein–protein interaction network for pKAL-regulated proteins using Cytoscape software. The results suggested that a significant survival network between TGF-β signaling proteins, insulin and NGF-β could be induced by pKAL.

### 2.4. TGF-β1 Inhibits pKAL-Induced Anticancer Effect

To elucidate the survival role of TGF-β1 in pKAL-induced cell death signaling, we examined the effect of TGF-β1 on pKAL-regulated cell morphology and viability by hematoxylin staining and CCK-8 assay. The results showed that pKAL-induced anticancer effects associated with morphological changes and downregulation of cell viability were significantly inhibited by co-treatment with TGF-β1, indicating a survival role of TGF-β1 under pKAL-induced cell death circumstance (Figure 5).

### 2.5. NGF-β Inhibits pKAL-Induced Anticancer Effect through PI3K Signaling

Phosphatidylinositide-3-kinase (PI3K) signaling is known to be involved in the survival role of insulin and NGF-β [16,17]. To elucidate the survival role of insulin and NGF-β in pKAL-induced cell death signaling, we examined the effects of insulin and NGF-β in the absence and presence of the PI3K inhibitors wortmannin and LY294002. The results showed that the ability of pKAL to reduce cell viability was significantly inhibited by NGF-β but not by insulin and this phenomenon was inhibited by wortmannin and LY294002 (Figure 6). These results suggested that growth factors and cytokines upregulated by pKAL may play a survival role under pKAL-induced cell death circumstance.

### 2.6. TGF-β1 and NGF-β Attenuate pKAL-Induced Cell Death through Upregulation of NF-κB p65 and Cyclin D1

NF-κB and cyclin D1 are known to be involved in the survival roles of TGF-β1 and NGF-β [18,19,20,21,22,23,24,25]. Western blot and densitometry analysis were performed to elucidate the survival mechanism induced by TGF-β1 and NGF-β during pKAL-induced cell death circumstance. As shown in Figure 7a,b, pKAL treatment alone induced upregulation of p53 and downregulation of caspase-8, NF-κB p65 and cyclin D1. As expected, downregulation of caspase-8, NF-κB p65 and cyclin D1 by 50 μg/mL pKAL was inhibited by co-treatment of either TGF-β1 or NGF-β with pKAL. These results indicated that TGF-β1 and NGF-β play a survival role in pKAL-induced cell death circumstance through upregulation of NF-κB p65 and cyclin D1.

## 3. Discussion

For successful cancer treatment, it is essential to study the survival mechanisms induced by chemotherapeutic anticancer drugs. Numerous in vitro experimental and in vivo clinical studies have been conducted to elucidate the anticancer mechanisms of natural dietary polyphenols. The aim of this study was to elucidate a novel mechanism for the more effective anticancer effect of polyphenols extracted from *Artemisia annua* L. (pKAL). We show here that certain survival signaling can occur under pKAL-induced cell death circumstance in p53-wild HCT116 colorectal cancer cells through upregulation of NF-κB and cyclin D1 by activation of TGF-β1 and NGF-β signaling pathways.

Extracellular vesicles (EVs) are heterogeneous cell-derived membrane structures including extracellular matrix proteins, growth factors, cytokines, chemokines, cell surface receptors, cytosolic signaling proteins, transcription factors, mRNAs, microRNAs, DNAs and lipids [26,27]. EVs are classified into two main categories, exosomes and microvesicles; exosomes are derived from the endosomal system of the intracellular region and secreted to the extracellular region and microvesicles are generated by shedding from the plasma membrane [28]. Many cell types, from embryonic stem cells to highly malignant cancer cells, secret different types of EVs and EVs derived from tumor cells are involved in cell-to-cell communication, cell proliferation, angiogenesis, invasion/migration, metastasis, immune modulation and drug resistance [26,29,30,31,32]. EVs can also be classified according to size as follows: small EVs (sEVs, 40–150 nm), medium EVs (mEVs, 150–1000 nm), apoptotic vesicles (ApoEVs, 100–1000 nm) and apoptotic bodies (1000–5000 nm) [28]. In this study, we showed that intracellular and extracellular vesicles were induced by pKAL during pKAL-induced cell death and morphological changes in HCT116 cells and the vesicles were more significantly detected by hematoxylin staining (Figure 1). These results suggest that pKAL-induced vesicles may contain nuclear-derived DNA, RNA, or acid nucleoprotein [33]. In addition, pKAL-induced vesicles detected by hematoxylin staining could be considered ApoEVs or apoptotic bodies due to their size and pKAL-induced apoptotic cell death. ApoEVs secreted from apoptotic glioblastoma cells are known to promote proliferation and therapy resistance of surviving tumor cells through intercellular transfer of splicing factors [34]. This fact indicates that pKAL-induced survival signaling during pKAL-induced cell death may be associated with pKAL-induced ApoEVs in HCT116 cells.

Colorectal cancer is one of the most common cancers with high incidence and mortality worldwide and originates from chronic inflammatory sites, cancer stem cells or stem cell-like cells [35,36]. A highly tumorigenic human colorectal cancer cell line (CR4) derived from liver metastasis is known to exhibit putative cancer stem cell characteristics [37]. In addition, stem cell-derived extracellular vesicles are involved in oncogenic processes [38]. Natural polyphenols are known to play a role in prevention and treatment of colorectal cancer due to their anti-inflammatory and antioxidant functions through inactivation of NF-κB signaling leading to the reduction of the secretion of inflammatory cytokines such as TNF-α, IL1β, IL-6 and IL-12 as well as inactivation of PI3K/AKT and β-catenin signaling leading to the inhibition of cell division [35,36]. Moreover, the immunomodulatory properties of polyphenols have been shown to be associated with their ability to modulate cytokine and chemokine production and activation of immune cells [35]. In this study, our results showed that various growth factors, cytokines and signaling mediators can be either upregulated or downregulated by pKAL in stem-like HCT116 cells (Table 1 and Table 2). Our studies suggest that various types of EVs induced by pKAL may be involved in upregulation of pKAL-induced growth factors and cytokines, resulting in pKAL-induced cell survival role.

The Epithelial to Mesenchymal Transition (EMT) process occurs in various types of epithelial cancers including colorectal cancer. TGF-β-induced EMT promotes the invasion and metastasis of colorectal cancer cells and this phenomenon is inhibited by resveratrol [39]. In addition, NGF, a neurotrophic factor, is known to be highly expressed in primary colorectal cancer tissues and promote colorectal cancer metastasis [40]. In this study, we revealed that pKAL-induced cell death was inhibited by TGF-β1 and NGF-β and this phenomenon was associated with upregulation of NF-κB p65 and cyclin D1 induced by TGF-β1 and NGF-β under pKAL-induced cell death circumstance (Figure 5, Figure 6 and Figure 7). Future studies of novel mechanisms to effectively inhibit the survival signaling induced by TGF-β1 and NGF-β signaling pathways may significantly contribute to novel therapies to overcome chemotherapeutic drug resistance in many patients suffering from various diseases, including cancer.

## 4. Materials and Methods

### 4.1. Materials

Penicillin-Streptomycin (10,000 U/mL) and TrypLE^TM^ Express Enzyme (1 X) with phenol red were from Thermo Fisher Scientific (Grand Island, NY, USA). RPMI 1640 medium was from HyClone (Logan, UT, USA). Dishes, plates, tubes and pipettes for cell culture were from SPL Life Sciences (Pocheon, Korea) or Thermo Fisher Scientific (Rockford, IL, USA). Cell counting kit-8 (CCK-8) was from Dojindo (Kumamoto, Japan). Hematoxylin solution and formaldehyde solution (4%) were from Merck KGaA (Darmstadt, Germany). Recombinant human TGF-β1 was from R&D Systems (Minneapolis, MN, USA). Human NGF-β, lipopolysaccharide (LPS, from *Escherichia coli* O111:B4), human insulin, wortmannin, LY294002 and phosphate buffered saline (PBS, pH 7.4) were from Sigma-Aldrich (St. Louis, MO, USA). Acrylamide/bis-acrylamide 37.5:1 solution (40%), Tween-20 and DMSO were from Amresco (Solon, OH, USA). Molecular weight marker (Precision Plus Protein Dual Color Standards) was from Bio-Rad (Hercules, CA, USA). Amersham Protran 0.2 μM NC membrane was from GE Healthcare Life Sciences. ECL Ottimo Western blot detection kit was from TransLab (Daejeon, Korea). X-ray film (CP-BU NEW) was from AGFA (Mortsel, Belgium). p53 (DO-1), NFκB-p65 (F-6), cyclin D1 (A-12) and GAPDH (FL-335) antibodies were from Santa Cruz (Santa Cruz, CA, USA). Caspase-8 (p18) antibody was from Invitrogen (Carlsbad, CA, USA). Secondary goat anti-rabbit and anti-mouse HRP conjugates were from Bio-Rad (Hercules, CA, USA) and Bethyl Laboratories (Montgomery, TX, USA), respectively.

### 4.2. pKAL Compounds

*Artemisia annua* L. polyphenols (pKAL) were extracted from the mixed tissues of *Artemisia annua* L. grown at Gaddongsook farm in Jinju, Korea by Prof. Sung Chul Shin using the previously described method [41]. To isolate pKAL compounds, mixed tissues including roots, stems, leaves and flowers of *Artemisia annua* L. were lyophilized, ground and extracted with 70% methanol at 60 °C for 20 h. The extract was filtered through a glass funnel and concentrated at 35 °C using a rotary evaporator. To remove fat components, the concentrated aqueous extract was extracted three times with equal volumes of *n*-hexane and methylene chloride. The filtrate was extracted three times with ethyl acetate to isolate the pKAL compounds and dried over anhydrous magnesium sulfate. pKAL compounds were identified by liquid chromatography-tandem mass spectrometry (LC/MS/MS) and quantified via HPLC-UV chromatogram analysis at 280 nm using eight standards in quintuplicate measurements (Figure 8, Table 3). For experiments, pKAL compounds were dissolved in DMSO solvent at a concentration of 100 mg/mL and stored in a −20 °C freezer until use.

### 4.3. Cell Culture

HCT116 human colorectal cancer cell line was purchased from Korean Cell Line Bank (KCLB No. 10247). HCT116 cells were cultured in RPMI medium with L-glutamine (300 mg/L), 25 mM HEPES, 25 mM NaHCO_3_, 1% penicillin/streptomycin and 10% heat inactivated FBS (Thermo Fisher Scientific, Grand Island, NY, USA). The cells were maintained and used for certain experiments after splitting every 3 days on a culture dish in a 37 °C incubator supplemented with 5% CO_2_ in a humidified atmosphere.

### 4.4. Cell Viability Analysis

Cells grown on a 24-well dish were incubated with maintenance medium containing 10% CCK-8 reagent for 1 h in a 37 °C CO_2_ incubator. The reaction solution (100 μL each) was then transferred to a 96-well dish and was analyzed by measuring the absorbance at OD_485 nm_ using a CHAMELEON microplate reader (Hidex, Turku, Finland).

### 4.5. Phase-Contrast Microscopy

Morphology of whole cells (attached and floating cells) was analyzed by phase-contrast microscopy (EVOS XL Core, Life Technologies) in a 10× objective (Inf Plan Achro 10× LWD PH, 0.25 NA/6.9 WD) with 150× amplification.

### 4.6. Phase-Contrast Microscopy of Hematoxylin-Stained Cells

Hematoxylin (cationic) staining is used to detect nucleus (DNA, RNA and acid nucleoprotein) [33]. Cells grown on a 6-well plate were washed with PBS, fixed with 4% formaldehyde solution, washed with PBS and then stained with hematoxylin solution for 30 min at RT. The cells were washed with PBS and were analyzed in the presence of 90% glycerol/PBS solution by phase-contrast microscopy (EVOS XL Core, Life Technologies) in a 20× objective (Inf Plan Fluor 20X LWD, 0.45 NA/7.1 WD) with 300× amplification.

### 4.7. Sample Preparation for Antibody Array

HCT116 cells were grown with 0.1% DMSO or 50 μg/mL pKAL for 60 h and collected by scraping in the presence of culture medium to harvest whole cells (attached and floating cells). After centrifugation, the cells were washed with PBS twice and the cell pellets were stored at −70 °C freezer until use. Proteins from the samples were extracted according to the RayBiotech sample preparation protocol and quantified with BCA protein assay kit (Pierce, Rockford, IL, USA).

### 4.8. Antibody Array Analysis

RayBiotech L-series human antibody array 1000 (Human L-1000) analysis was performed by the ebiogen company (Seoul, Korea). The antibody array results simultaneously show the expression levels of 1000 human proteins including cytokines, chemokines, adipokine, growth factors, angiogenic factors, proteases, soluble receptors, soluble adhesion molecules and other proteins. The experimental procedure using Human L-1000 is briefly described as follows. Antibody array slide (RayBiotech, Norcross, GA, USA) was dried and incubated with blocking solution for 30 min. After decanting the blocking buffer, the slide was incubated with 400 μL of diluted sample for 2 h at RT. After decanting the samples, each array slide was washed three times with 800 μL of wash buffer with shaking. The slide was incubated with biotin-conjugated anti-cytokine antibodies for 2 h at RT with gentle shaking, washed, incubated with Cy3-Conjugated Streptavidin for 2 h at RT, washed, rinsed with deionized water and centrifuged at 1000 rpm for 3 min to remove water. The slide scanning was performed using GenePix 4100 A Scanner (Axon Instrument, Scottsdale, AZ, USA) and the signals were quantified with GenePix Software (Axon Instrument, Scottsdale, AZ, USA), After analyzing, the data about protein information was annotated using UniProt DB.

### 4.9. Bioinformatics Analysis

Significant genes corresponding to 10 upregulated and 13 downregulated proteins by pKAL more than 1.4-fold were analyzed for expression heatmap, keywords and gene category classification using Excel based Differentially Expressed Gene Analysis (ExDEGA) software (v.3.0.0) provided by ebiogen company and the Database for Annotation, Visualization and Integrated Discovery (DAVID) program (v.6.8) through the internet (http://david.ncifcrf.gov/tools.jsp) (accessed on 1 October 2021). String analysis for the protein–protein interaction network was performed using Cytoscape software (v.3.7.0.) over the internet (http://www.cytoscape.org) (accessed on 1 October 2021). In addition, functional information of the pKAL-regulated proteins was analyzed using the UniProt website (http://uniprot.org) (accessed on 1 October 2021).

### 4.10. Western Blot Analysis

Whole cells (attached and floating cells) were extracted with SDS sample buffer and were boiled for 5 min at 95 °C. The resultant proteins were separated using SDS-PAGE and transferred to an NC membrane at 30 mA for 13–15 h. After washing with PBST (0.1% Tween-20, PBS) twice for 1 h, the membrane was blocked for 30 min at RT in blocking buffer (3% skim milk, 0.1% Tween-20, PBS) and then incubated with primary antibody in blocking buffer at 4 °C overnight. The blot was then washed with PBST three times for 10 min and incubated with an HRP-conjugated secondary antibody in blocking buffer for 2 h at RT. After being washed with PBST, the blot was analyzed with the ECL Western blot detection system.

### 4.11. Statistical Analysis

Results for cell viability are presented as mean ± standard deviation of the mean. Statistical significance between control and sample was determined using Student’s *t*-test. Values of *p* < 0.05 are considered statistically significant.

## Figures and Tables

**Figure 1 ijms-23-01598-f001:**
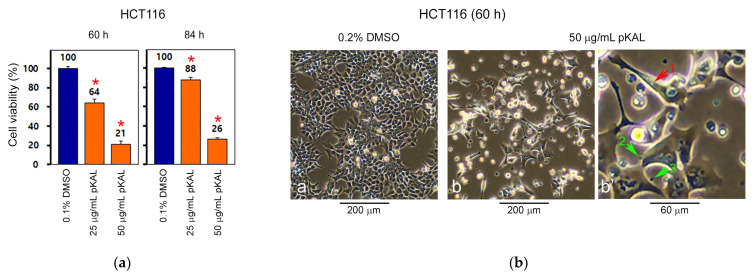
Long-term effects of pKAL on cell viability and morphology in p53-wild HCT116 colorectal cancer cells. HCT116 cells were grown with DMSO or pKAL for 60 and 84 h: (**a**) Cell viability was analyzed by CCK-8 assay in triplicate tests. Error bars represent standard deviation of the mean. Statistical significance between control and sample was determined using Student’s *t*-test, * *p* < 0.05; (**b**) Phase-contrast microscopy of whole cells; (**c**) Phase-contrast microscopy of hematoxylin-stained attached cells. Panel b’ in Figure 1b,c was enlarged from panel b.

**Figure 2 ijms-23-01598-f002:**
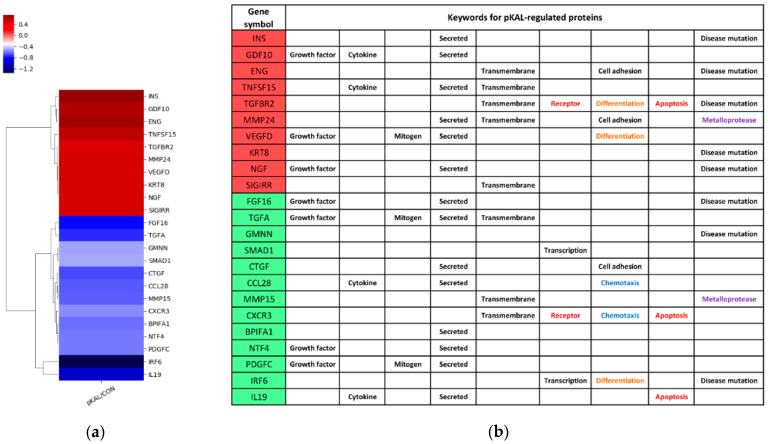
Bioinformatics analysis for pKAL-regulated proteins: (**a**) Expression heatmap analysis; (**b**) Keywords analysis.

**Figure 3 ijms-23-01598-f003:**
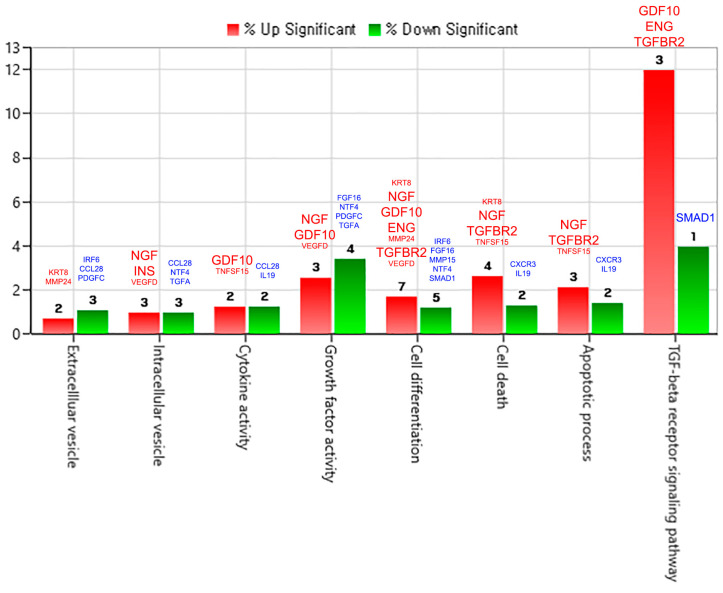
Gene category chart for pKAL-regulated proteins.

**Figure 4 ijms-23-01598-f004:**
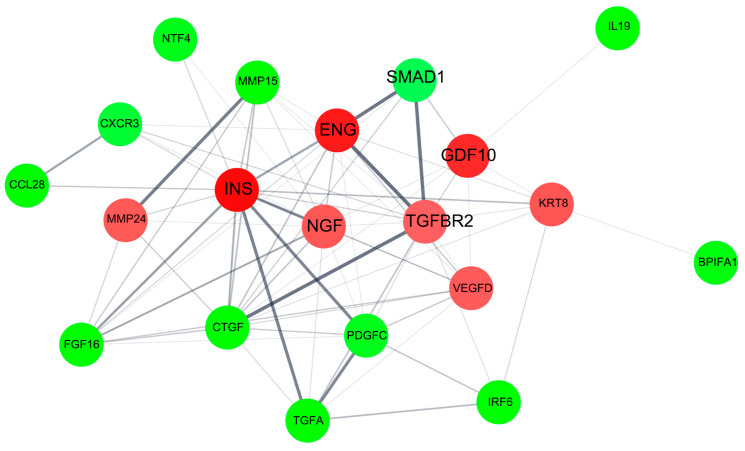
Protein–protein interaction network for pKAL-regulated proteins. Red indicates the upregulated proteins and green indicates the downregulated proteins by pKAL.

**Figure 5 ijms-23-01598-f005:**
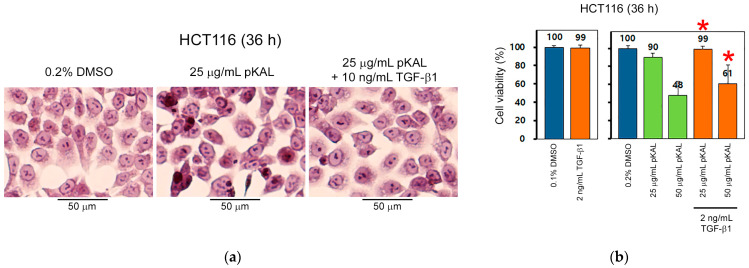
Effect of TGF-β1 on pKAL-regulated morphological changes and cell viability. HCT116 cells were grown for 16 h and then treated with the indicated amounts of DMSO, pKAL and TGF-β for 36 h: (**a**) Hematoxylin staining; (**b**) Cell viability was analyzed by CCK-8 assay in triplicate tests. Statistical significance between control and sample was determined using Student’s *t*-test, * *p* < 0.05.

**Figure 6 ijms-23-01598-f006:**
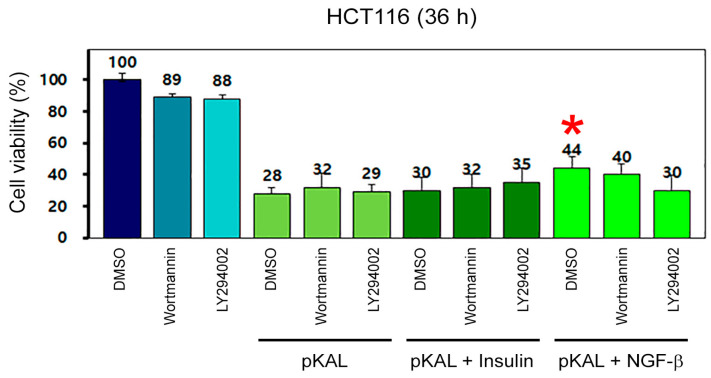
Effects of PI3K inhibitors, insulin and NGF-β on pKAL-regulated cell viability. HCT116 cells were grown for 20 h and then treated with the indicated combinations of 0.1% DMSO, 0.5 μM wortmannin, 0.5 μM LY294002, 50 μg/mL pKAL, 200 nM insulin and 100 ng/mL NGF-β for 36 h. Two independent experiments for cell viability analysis were performed in triplicate tests. Statistical significance between control and sample was determined using Student’s *t*-test, * *p* < 0.05.

**Figure 7 ijms-23-01598-f007:**
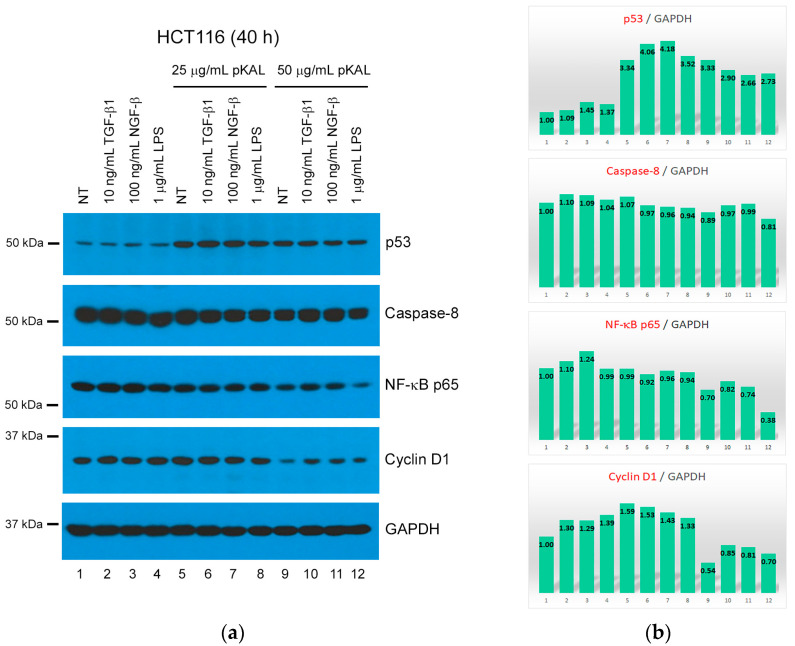
Effects of TGF-β1, NGF-β and LPS on pKAL-induced cell death mechanism. HCT116 cells were grown for 20 h and then treated with the indicated amounts of drugs for 40 h: (**a**) Western blot analysis using the indicated antibodies; (**b**) Densitometry analysis of protein bands in the panels of Figure 7a.

**Figure 8 ijms-23-01598-f008:**
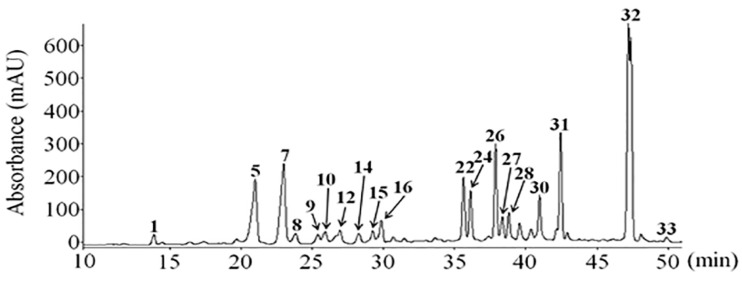
HPLC chromatogram of pKAL compounds extracted from mixed tissues including roots, stems, leaves and flowers of *Artemisia annua* L: Peak **1**, Caffeic acid; **5**, Quercetin-3-O-galactoside; **7**, Mearnsetin-glucoside; **8**, Kaempferol-3-O-glucoside; **9**, Quercetin-3-O-glucoside; **10**, Mearnsetin-glucoside; **12**, Ferulic acid; **14**, Isorhamnetin-glucoside; **15**, Diosmetin-7-O-d-glucoside; **16**, Luteolin-7-O-glucoside; **22**, Quercetin; **24**, Quercetagetin-3-O-methyl ether; **26**, Luteolin; **27**, 8-methoxy-kaempferol; **28**, Quercetagetin-5,3-di-O-methyl ether; **30**, Kaempferol; **31**, 3,5-dihydroxy-6,7,4′-trimethoxyflavone; **32**, 3,5-dihydroxy-6,7,3′,4′-tetramethoxyflavone; **33**, Isorhamnetin.

**Table 1 ijms-23-01598-t001:** Identification of growth factors, cytokines and mediators upregulated by pKAL through antibody array.

Gene Symbol	Upregulated Proteins by pKAL	Fold Change pKAL/CON	Normalized Control	Normalized pKAL	Mass (Da)	UniProt ID
INS	Insulin	1.677	874.2	1466.3	11,981	P01308
ENG	Endoglin	1.62	553	895.9	70,578	P17813
GDF10	Growth/differentiation factor 10	1.573	528.7	831.7	53,122	P55107
TNFSF15	Tumor necrosis factor ligand superfamily member 15	1.512	592	895.4	28,087	O95150
	Keratin, type II cytoskeletal 8					
KRT8	Single Ig IL-1-related receptor	1.438	581.8	836.8	53,704	P05787
SIGIRR	Beta-nerve growth factor	1.434	524.2	751.7	45,679	Q6IA17
NGF	Matrix metalloproteinase-24	1.43	440.5	630	26,959	P01138
MMP24	Vascular endothelial growth factor D	1.425	703.6	1002.7	73,231	Q9Y5R2
VEGFD	TGF-beta receptor type-2	1.42	499	708.5	40,444	O43915
TGFBR2		1.407	523.8	736.8	64,568	P37173

**Table 2 ijms-23-01598-t002:** Identification of growth factors, cytokines and mediators downregulated by pKAL through antibody array.

Gene Symbol	Downregulated Proteins by pKAL	Fold Change pKAL/CON	Normalized Control	Normalized pKAL	Mass (Da)	UniProt ID
IRF6	Interferon regulatory factor 6	0.391	2311.8	904.6	53,130	O14896
IL19	Interleukin-19	0.504	1715.2	865.1	20,452	Q9UHD0
FGF16	Fibroblast growth factor 16	0.573	2972.8	1703	23,759	O43320
TGFA	Protransforming growth factor alpha	0.602	920	554	17,006	P01135
CTGF	Connective tissue growth factor	0.625	778.2	486.2	38,091	P29279
CCL28	C-C motif chemokine 28	0.638	6431.9	4104.8	14,280	Q9NRJ3
MMP15	Matrix metalloproteinase-15	0.643	745	479	75,807	P51511
BPIFA1	BPI fold-containing family A member 1	0.66	1417	935.4	26,713	Q9NP55
NTF4	Neurotrophin-4	0.665	963.3	640.7	22,427	P34130
PDGFC	Platelet-derived growth factor C	0.666	1023.3	681.3	39,029	Q9NRA1
CXCR3	C-X-C chemokine receptor type 3	0.685	893.2	611.5	40,660	P49682
GMNN	Geminin	0.709	2321.9	1646.7	23,565	O75496
SMAD1	Mothers against decapentaplegic homolog 1	0.713	752.8	537	52,260	Q15797

**Table 3 ijms-23-01598-t003:** Mass spectra of pKAL compounds identified by LC/MS/MS.

Compounds	[M − H]^−^	Fragments (*m/z*)
Caffeic acid (**1**)	179	179 135
Quercetin-3-O-galactoside (**5**)	463	463 301 175 151 121
Mearnsetin-glucoside (**7**)	493	493 331 315 287 270 181
Kaempferol-3-O-glucoside (**8**)	447	447 285
Quercetin-3-O-glucoside (**9**)	463	301 300 178 175 151 121 107
Mearnsetin-glucoside (**10**)	493	493 331 315 287 270 181
Ferulic acid (**12**)	193	193 178 161 149 134
Isorhamnetin-glucoside (**14**)	477	477 462 446 315 314 313 300 299 287 271
Diosmetin-7-O-d-glucoside (**15**)	461	461 341 299
Luteolin-7-O-glucoside (**16**)	447	447 285 255 227
Quercetin (**22**)	301	301 273 179 151 121 107
Quercetagetin-3-O-methyl ether (**24**)	331	331 316 287 271 209 151 179 166
Luteolin (**26**)	285	285 243 241 217 198 175 151 133
8-methoxy-kaempferol (**27**)	315	315 300 285 137
Quercetagetin-5,3-di-O-methyl ether (**28**)	345	345 330 315 287
Kaempferol (**30**)	285	285 217 151 133
3,5-dihydroxy-6,7,4′-trimethoxyflavone (**31**)	359	359 344 329 314 301 286
3,5-dihydroxy-6,7,3′,4′-tetramethoxyflavone (**32**)	373	373 358 343 328 315 299 285
Isorhamnetin (**33**)	315	315 300 271 247 203

## Data Availability

Not applicable.

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
