# Peer review of "Identification of Growth Factors, Cytokines and Mediators Regulated by Artemisia annua L. Polyphenols (pKAL) in HCT116 Colorectal Cancer Cells: TGF-β1 and NGF-β Attenuate pKAL-Induced Anticancer Effects via NF-κB p65 Upregulation"

_ijms, 2022, doi:10.3390/ijms23031598_

Round 1

Reviewer 1 Report

I value the article submitted for review and recommend it for publication in IJMS. The submitted manuscript " Identification of Growth Factors, Cytokines and Mediators Regulated by Artemisia annua L. Polyphenols (pKAL) in HCT116 Colorectal Cancer Cells: TGF-b1 and NGF-b Attenuate pKAL-induced Anticancer Effects via NF-kB p65 Upregulation " provides important data. The work is interesting, transparent and well organized. The introduction provides a sufficient background. I just have a few small comments regarding the description of the methods that are mentioned in the following lines.

-line 310 (4.2. pKAL components) This section gives information that pKAL was isolated from all Artemisia annua tissue and that pKAL analysis was performed as previously described [41]. However, this reference 41 says nothing about pKAL, but only about other extracts. Please add information about the process of obtaining the test fraction (pKAL) and its mass spectrometry analysis or provide appropriate references.

- A subsection of the “statistical analysis” is missing in the "method" section. It would be appropriate to write how many replications of the test samples were made in all experiments and describe how the statistical analysis was performed.

Reviewer 2 Report

The paper submitted by Jung and coworkers is focused on the evaluation of the molecular pathways of action of pKAL obtained from Artemisia annual. The bioassays presented in the manuscript are properly performed and described and obtained results are interesting for the scientific community. However,  the paper lacks proper phytochemical analysis of the pKAL and without adding additional information it should not be published in IJMS.

1) it is not clear what is whole tissues in reference to the plant material - in cited paper 10.1002/bmc.3587 the phytochemical analysis does not refer to whole tissue

2) was pKAL analyzed in 10.1002/bmc.3587? is so please clearly state that this is the same extract as in the cited paper.

3) is section 4.2 the authors state that the pKAL "were isolated" - please explain exactly how it was isolated. the extract cannot be isolated; usually isolation refer to pure compound or specific fraction obtained by chromatography 

4) the plant material used fro bioassay should be correctly described - please state where it was grown, voucher specimen of the material and pKAL should be deposited, authentication of the plant material should be performed according to one of standard procedures, name of the person responsible for the authentication should be provided

5) results of phytochemical screening must be provided unless this is exactly the same material that was used in previous studies and the phytochemical analysis was already done and described in the proper manner

Round 2

Reviewer 2 Report

The paper was significantly improved and can be accepted for publication in IJMS.